# Aqueous-Based Synthesis of Photocatalytic Copper Sulfide Using Sulfur Waste as Sulfurizing Agent

**DOI:** 10.3390/ma15155253

**Published:** 2022-07-29

**Authors:** Gabriele Sarapajevaite, Davide Morselli, Kestutis Baltakys

**Affiliations:** 1Department of Silicate Technology, Kaunas University of Technology, Radvilenu Road 19, LT-50254 Kaunas, Lithuania; kestutis.baltakys@ktu.lt; 2Department of Civil, Chemical, Environmental, and Materials Engineering, Università di Bologna, Via Terracini 28, 40131 Bologna, Italy; davide.morselli6@unibo.it; 3National Interuniversity Consortium of Materials Science and Technology (INSTM), Via Giusti 9, 50121 Firenze, Italy

**Keywords:** CuS, photocatalysts, dye degradation, hydrothermal synthesis, waste valorization

## Abstract

Most of the copper sulfide synthetic approaches developed until now are still facing issues in their procedure, such as long synthesis duration, high energetic consumption, and high implementation costs. This publication reports a facile and sustainable approach for synthesizing copper sulfides on a large scale. In particular, an industrial by-product of sulfur waste was used as a sulfurizing agent for copper sulfide synthesis in a water medium. The reaction was performed in the hydrothermal environment by following a novel proposed mechanism of copper sulfide formation. The investigation of morphological and optical properties revealed that the target products obtained by using waste possess the resembling properties as the ones synthesized from the most conventional sulfurizing agent. Since the determined band gap of synthesis products varied from 1.72 to 1.81 eV, the photocatalytic properties, triggered under visible light irradiation, were also investigated by degrading the methylene blue as a model pollutant. Importantly, the degradation efficiency of the copper sulfide synthesized from sulfur waste was equivalent to a sample obtained from a reference sulfurizing agent since the value for both samples was 96% in 180 min. This very simple synthetic approach opens up a new way for large-scale sustainable production of visible-light-driven photocatalysts for water purification from organic pollutants.

## 1. Introduction

Over the last few decades, constant research for novel materials fabrication methods applied in sustainable technologies has been in high demand. In recent years, semiconductors, especially transition metal chalcogenides, revealed great potential for application in the fabrication of materials used for renewable energy devices [1,2,3,4,5]. Compared to other transition metal chalcogenides, copper sulfide (Cu_x_S, x = 1–2) is considered to be an abundant, low-cost compound, which has desirable properties for p-type semiconductors, such as high theoretical specific capacity (560 mAh/g), appropriate direct band gaps for solar light absorption, potential high carrier concentrations, low thermal conductivity, plasmonic properties, etc. [1,6,7,8,9]. Copper sulfide also shows eligible properties for biomedical [10], antibacterial [11,12], photocatalytic [13,14], and thermoelectric [15] applications. For these reasons, copper sulfide could potentially be used as a material for the production of light-emitting diodes [16], researchable batteries [17,18,19], solar cells [4,7,20,21,22], photoelectronic devices [23], supercapacitors [2,24,25], biosensors [26], DNA detectors [27] etc. Moreover, copper sulfides showed outstanding results in various photodegrading dyes under UV or visible light irradiation [28,29].

Copper sulfide exists in different stoichiometries, and thermodynamically stable compounds at room temperature are the following: chalcocite (Cu_2_S), djurleite (Cu_1.96_S), digenite (Cu_1.85_S), anilite (Cu_1.75_S) and covellite (CuS) [1]. A considerable amount of literature has been published on approaches to fabricating copper sulfides. According to these studies, chemical vapour deposition [30], chemical bath deposition [31], simple chemical precipitation [8,21,32], high temperature thermolysis [33], electrochemical synthesis [34], sonochemistry [16], mechanochemistry [12,35,36], sonication [37], solvothermal method [29,38] have been developed over the past few decades. Among them, hydrothermal synthesis is a widely employed technique due to process advantages, such as low production cost, easy processing, and eco-friendliness [39,40,41,42].

Although the investigation of copper sulfides synthesis covers several decades, among the diverse developed methodology to control the morphology of the products, researchers still face issues such as long synthesis duration and high energy consumption [25,39,43,44]. Furthermore, the majority of copper sulfide synthesis is carried out in organic reagents as a reaction medium, which is environmentally unsustainable and human hazardous, such as oleyl amine, hydrazine hydrate, or fossil-based, e.g., polyvinylpyrrolidone or toluene [44,45]. Moreover, in most cases, the synthesis is carried out on a very small scale and with yields of the target product quite low, which are unsuitable if a scaleup of the process is envisioned [44]. In addition, copper sulfide production costs are typically very high; thus, the valorization of a sulfuring agent can reduce the production costs. The use of a by-product coming from other products as a sulfuring agent can further reduce the production costs of copper sulfide.

Data from several studies suggested that compounds that release copper (e.g., CuCl_2_·2H_2_O, CuSO_4_·5H_2_O, Cu(NO)_3_·3H_2_O) and sulfur (e.g., Na_2_S·9H_2_O, Na_2_S_2_O_3_, CH_4_N_2_S) elements in an ionic state, were usually applied for typical copper sulfide synthesis since the cation exchange reaction is one of the most explored metal sulfide synthesis method [45,46,47]. However, few studies have investigated the application of cheap and earth-abundant reagents, for instance, insoluble copper or sulfur sources [14,48]. When taking into consideration the aforementioned environmental aspects, the investigation of a new approach for copper sulfides synthesis under sustainable conditions has now become a necessity. For example, J. Kundu and Pradhan successfully synthesized CuS microspheres in a water medium [49]. Nevertheless, the efforts to investigate the sustainable approach to synthesizing a high-value product are low since most of the proposed synthetic pathways are based on low yield production in a toxic reagents environment.

This study reports the facile, environmentally friendly, and mild conditions approach for synthesizing copper sulfides on a large scale and explores a new mechanism of the synthesis reaction. Moreover, the purpose of this research is to synthesize copper sulfide in the aqueous solution by using it as a sulfurizing agent from sulfur waste derived from sulfuric acid production. The morphology and photocatalytic performance of target products are also investigated, using methylene blue as a model contaminant. Specifically, the analyzed synthesis method proposes the possibility of producing copper sulfides in a non-hazardous water medium without the need for product purification. The hydrothermal treatment enables high-yield material fabrication without complex technological apparatus. Since the industrial waste was applied as sulfur raw material, the suggested approach is not only sustainable but also reduces the price of fabrication.

## 2. Materials and Methods

### 2.1. Materials

Sulfur waste was used as a sulfurizing agent. The waste was formed in a chemical plant “Lifosa” (Kedainiai, Lithuania), by producing sulfuric acid. The by-product was obtained from filters of melted elemental sulfur. According to X-ray fluorescence analysis (XRF), the waste consisted of sulfur (86.3%), calcium (6.9%), iron (4%), and silicon (2.3%). Before the experiments, sulfur waste was milled in a vibrating cup for 2 min at 900 rpm.

As a reference, elemental sulfur (99.9%) was also used for the experiments. Powder of copper oxide (99%, Reachem, Slovakia) was used as the copper source.

### 2.2. Synthesis of Copper Sulfide

The experiment mixture was prepared by mixing sulfur waste with copper oxide with molar ratio of S and CuO equal to 1:1.5. Both materials were mixed in homogenizer (Turbula Type T2F, Willy A Bachofen AG, Muttenz, Switzerland) for 1 h at 34 rpm and further followed by milling 10 g of mixture in a 50 mL planetary mill “Pulverisette 9” for 1 min at 600 rpm speed.

Furthermore, the mixture with sulfur reference as sulfur source was also prepared using the same material preparation conditions deliberately to validate if sulfur waste is applicable raw material for copper sulfide preparation.

Hydrothermal synthesis was performed on a stainless steel autoclave (“Parr instruments”, Frankfurt am Main, Germany) in a saturated steam environment. The sample treatment was carried out by stirring 5 g of primary mixture with 50 mL of water at 50 rpm rate in a 150 mL vessel. The reaction temperature was 145 or 180 °C, which was reached by increasing 100 °C per hour. The isothermal treatment was performed for 0.5 or 4 h (Table 1). After the isothermal treatment, the reaction vessel was cooled down to room temperature naturally. Then the sample suspension was filtered, and the pH of liquid phase was measured. Black precipitate was dried under air atmosphere at 104 °C temperature for 12 h, grinded, sieved through 100 μm sieve, and analyzed by instrumental analysis.

### 2.3. Thermodynamic Calculations

For the thermodynamic calculations of the hypothetical reaction parameters, a method of absolute entropies was used [50,51,52]. According to this method, the change of reaction standard free Gibbs energy $\Delta_{r}G_{T}^{0}$ was calculated by the equation:(1)$$\Delta_{r}G_{T}^{0}=\Delta_{r}H_{T}^{0}-T\Delta_{r}S_{T}^{0}$$
where: $\Delta_{r}H_{298}^{0,f}$, and $\Delta_{r}S_{298}^{0,f}$—the change/alteration/variation of reaction enthalpy and entropy in temperature *T*. The used standard molar thermodynamic properties values at 25 °C and 1 bar are given in Table 2 [53,54,55].

### 2.4. Photocatalytic Activity Experiment

The evaluation of photocatalytic activity was carried out by photodegrading the aqueous solution of methylene blue (MB) (20 mg‧L^−1^) under visible light irradiation. Sw 0.5 h 180 °C and Sr 0.5 h 180 °C samples were chosen as photocatalysts for this experiment. A total of 15 mg of synthesized copper sulfide were mixed in a vessel with 100 mL of MB solution and 4 mL of H_2_O_2_ according to parameters optimized in a previous study [29]. In order to achieve an adsorption–desorption equilibrium, the catalyst was magnetically stirred in a dye solution for 10 min before starting the irradiation. The suspension was then exposed to visible light by 23 W LED light bulb at room temperature with constant stirring. The whole irradiation lasted for 180 min. After specified irradiation intervals, an aliquot of MB solution (5 mL) was sampled and centrifuged for 5 min to separate the particles of the photocatalyst. The UV-Vis spectrum of the supernatant was measured by spectrophotometer (PerkinElmer Lambda 25 UV/VIS to determine the characteristic absorption peak centered at 664 nm and to estimate the concentration of MB. The degradation efficiency was calculated by following formula:(2)$$\eta =\left(1-\frac{c_{A}}{c_{A0}}\right)\cdot 100\%\;$$
where *η*—degradation efficiency; *c_A_*—the concentration of solution at time t; *c_A_*_0_—the primary concentration of methylene blue solution (mg‧L^−1^). The degradation rate was determined according to equation of pseudo-first-order kinetics model (Equation (3)) [29].
(3)$$ln\frac{c_{A}}{c_{A0}}=-kt$$
*k*—the reaction rate constant (min^−1^).

Tests in the dark with or without photocatalysts and with or without hydrogen peroxide were also performed for comparison reasons. The results were presented in supporting information (Appendix A).

### 2.5. Characterization Methods

X-ray fluorescence spectroscopy was applied to determine the quantitative chemical composition of applied waste by using Bruker X-ray S8 Tiger WD spectrometer. The detailed experimental conditions are described in our previous work [56]. The X-ray diffraction analysis was used to identify the composition of chemical compounds by using D8 Advance diffractometer (Bruker AXS, Karlsruhe, Germany). Ni 0.02 nn filter was used for filtration of X-ray beam in order to select CuKα wavelength. The chosen scanning range was 2Θ = 2–70°, which was obtained by 6 °min^−1^ scanning speed. The detailed description of performed measurements are also characterized in our previous publication [56]. XRD software “Diffract.Eva” was applied to estimate the intensity of diffraction maximums and crystallite size of samples. For this reason, five main diffraction maximums of hexagonal covellite (PDF 04-006-9635) were investigated: 0.305 nm at 29.250° [102], 0.282 nm at 31.757° [103], 0.273 nm at 32.820° [006], 0.190 nm at 47.889° [110], 0.174 nm at 52.669° [108].

The particle size distribution of synthesized products was determined by using CILAS 1090 LD grain-size analyzer, which has a sensitivity range from 0.04 to 500 μm. The distribution of solid particles in water medium was 12–15%. The measuring time was 60 s.

The optical properties of synthesized material were analyzed by performing diffuse reflectance measurements in the range of 200–900 nm using Jasco V-650 UV-vis-NIR spectrophotometer calibrated with BaSO_4_ reference before starting measurements. The band gap of copper sulfide was estimated by applying the equation of Kubelka–Munk theory [57].

Morphology and chemical composition of samples were investigated in a dual beam system FE-SEM-FIB Helios Nanolab 650 (FEI Company, Fei Europe B.v., Eindhoven, The Netherlands) equipped with EDX spectrometer X-Max (Oxford Instruments, Paris, France). The transmission electron microscope (TEM) images were acquired by Tecnai G2 F20 X-TWIN spectrometer.

## 3. Results

### 3.1. Products of Hydrothermal Synthesis

In order to establish the path of copper sulfide formation from elemental sulfur and copper oxide in a water medium, thermodynamic calculations were performed. According to thermodynamic estimations, during hydrothermal synthesis, the melted liquid sulfur firstly reacted with water molecules to produce hydrogen sulfide gas and sulfur dioxide in a soluble state (Equation (4)). The latter equation is also known as the reverse Claus reaction, which according to the literature, is able to occur in mixtures containing metal sulfides at temperatures under 200 °C [58]. Moreover, the produced hydrogen sulfide further reacts with particles of copper oxide by forming copper sulfide (Equation (5)). Meanwhile, the produced sulfur dioxide reacts with water by disproportion reactions to produce hydrogen cations (Equations (6) and (9)), which further reacts with melted sulfur and copper oxide (Equations (7) and (8)) [59]. Synthesized H_2_S (through Equation (7)) was used for reaction with CuO, whereas copper cations were formed by Equation (8). The latter constituent further reacted with HS^−^, which is a product of the second disproportion reaction, to produce the target synthesis product—covellite (Equation (10)). According to the proposed mechanism, the sample solution consisted of H_2_O, SO_4_H^−^, H^+^, and the other possibly unreacted compounds, such as H_2_S, SO_2_, HSO_3_^−^, and HS^−^. It is worth mentioning that the estimated amount of free Gibb’s energy of Equations (5), (7), and (10) were far below the zero point; therefore, the suggested mechanism is highly reliable (Figure 1). The proposed theoretical reactions:3S (l) + 2H_2_O (l)→SO_2_ (aq) + H_2_S (g)(4)
CuO (s) + H_2_S (g)→CuS (s) + H_2_O (l)(5)
SO_2_ (aq) + H_2_O (l)↔H_2_SO_3_ (aq)↔H^+^ (aq) + HSO_3_^−^ (aq)(6)
S (l) + H^+^ (aq)→H_2_S (g)(7)
CuO (s) + H^+^ (aq)→Cu^2+^ (aq) + H_2_O (l)(8)
S (s) + HSO_3_^−^ (aq) + H_2_O (l)↔S_2_O_3_H^−^ (aq) +H_2_O (l)↔HS^−^ (aq) + SO_4_H^−^ (aq)(9)
Cu^2+^ (aq) + HS^−^ (aq)→CuS (s) + H^+^ (aq)(10)

In addition, during the hydrothermal treatment, elemental sulfur was the raw material in the formation of soluble sulfur compounds. The formed anions and cations further reacted with copper oxide in producing copper sulfide particles.

Consequently, to prove theoretical estimations, the mineralogical composition of synthesis samples was investigated. It was observed that hexagonal (P63/mmc) CuS phase—covellite (PDF 04-006-9635)—(d—*spacing—* 0.305; 0.281; 0.273 nm) was synthesized by 0.5 h hydrothermal treatment at 145 °C using both samples with sulfur waste and reference. Furthermore, the covellite was the main constituent of samples by increasing synthesis temperature from 145 to 180 °C and by increasing duration up to 4 h (Figure 2). The origin components of sulfur waste, such as quartz (d—*spacing—*0.428; 0.335 nm) and anhydrite (d—*spacing—*0.350 nm), were identified in all sulfur waste samples; thus, these compounds remain unreacted through all synthesis conditions (Figure 2A). By comparing samples with sulfur waste treated under different conditions, the unreacted sulfur (d—*spacing*—0.386; 0.345 nm) was only noticed in the Sw145C0.5h sample (Figure 2A). Meanwhile, the latter constituent was identified in reference samples obtained during all experimental conditions (Figure 2A). The Sr145C0.5h sample also consisted of a primary mixture compound—tenorite (CuO, d—*spacing—*0.253; 0.233 nm) (Figure 2B). By comparing samples treated in 145 °C 0.5 h conditions, the higher intensity diffraction maximums of CuO and S were identified in the reference sample than in the waste sample, and this has indicated that the formation of CuS was slower in a reference sample than in waste. This phenomenon was explained due to the reactivity differences in sulfurizing agents because of the diverse mechanical preparation of both sulfurizing materials. According to the obtained results, the sulfurizing agent affects the formation of minor and secondary products, whereas the formation of the target synthesis product was affected insignificantly.

Consequently, to prove theoretical estimations, the mineralogical composition of synthesis samples was investigated. It was observed that hexagonal (P63/mmc) CuS phase—covellite (PDF 04-006-9635)—(d—*spacing—* 0.305; 0.281; 0.273 nm) was synthesized by 0.5 h hydrothermal treatment at 145 °C using both samples with sulfur waste and reference. Furthermore, the covellite was the main constituent of samples by increasing synthesis temperature from 145 to 180 °C and by increasing duration up to 4 h (Figure 2). The origin components of sulfur waste, such as quartz (d—*spacing—*0.428; 0.335 nm) and anhydrite (d—*spacing—*0.350 nm), were identified in all sulfur waste samples; thus, these compounds remain unreacted through all synthesis conditions (Figure 2A). By comparing samples with sulfur waste treated under different conditions, the unreacted sulfur (d—*spacing*—0.386; 0.345 nm) was only noticed in the Sw145C0.5h sample (Figure 2A). Meanwhile, the latter constituent was identified in reference samples obtained during all experimental conditions (Figure 2A). The Sr145C0.5h sample also consisted of a primary mixture compound—tenorite (CuO, d—*spacing—*0.253; 0.233 nm) (Figure 2B). By comparing samples treated in 145 °C 0.5 h conditions, the higher intensity diffraction maximums of CuO and S were identified in the reference sample than in the waste sample, and this has indicated that the formation of CuS was slower in a reference sample than in waste. This phenomenon was explained due to the reactivity differences in sulfurizing agents because of the diverse mechanical preparation of both sulfurizing materials. According to the obtained results, the sulfurizing agent affects the formation of minor and secondary products, whereas the formation of the target synthesis product was affected insignificantly.

The particle size of bulk synthesized material was further investigated. It was determined that mainly all the samples consisted of particles less than 100 μm (Figure 3). However, the dominant particle size of samples was less than 30 μm because 53.7–70.1% of bulk material consisted of the latter value. Furthermore, the particles less than 1 μm consisted of 5.9–9.9% of the samples (Figure 3). It is worth noting that the largest particle size was characteristic of the Sr 0.5h 145 °C sample (Figure 3). According to XRD analysis, the Sr 0.5 h 145 °C sample contained the highest content of unreacted sulfur (Figure 2). In addition, during the cooling stage of hydrothermal treatment, the melted sulfur solidified into particles that were bigger than synthesis products and resulted in an increase in the average particle size (Figure 3).

However, the particle size of bulk material is commonly influenced by treatment conditions, such as processing medium and drying conditions; on the contrary, the crystallite size is a parameter that is affected by the functional properties of semiconductor material [60]. It was determined that the smallest crystallites of covellite were obtained during synthesis at 145 °C. The size of the waste and reference samples, respectively, was 33 nm and 27 nm on average (Table 3). By increasing the hydrothermal treatment temperature up to 180 °C, the crystallite size increases by 6 nm in waste and 8 nm in reference samples, showing that the growth of crystals accelerates with the reaction temperature (Table 3). On the other hand, the synthesis time did not significantly affect the average crystallite size, which increased by 1–2 nm, increasing the reaction time from 0.5 to 4 h (Table 3). Furthermore, we have noticed that the sulfurizing agent slightly affected the size of crystallites since the values of the latter parameter were systematically lower, for the reference sample, by 2–5 nm for each experimental point (Table 3). Therefore, the temperature of hydrothermal treatment was the main parameter stimulating the growth of crystals.

According to SEM images, the waste samples exhibited round shape plate morphology, and by increasing the temperature, the particles conglomerated into flower-shaped clusters (Figure 4.). TEM images also confirmed the tabular structure synthesis products (Figure 4). Furthermore, waste samples were investigated by energy dispersive X-ray spectroscopy (Table 4), which confirmed that the synthesized Cu_x_S compound was covellite because the ratio of S to Cu in Sw0.5 h 145 °C and Sw0.5 h 180 °C samples were 1:1 (Table 4). Although the Cu:S ratio in the Sw4 h 180 °C sample was 0.9:1, which indicated the formation of sulfur-rich copper sulfide compound, according to XRD results, the latter compound was not indicated; therefore, the obtained results could be due to the instrumentation error.

### 3.2. Photocatalytic Effect

Photocatalytic properties are characteristic of copper sulfide since the latter material generates free electrons when irradiated by an electromagnetic wave [45]. To identify the wavelength that stimulates the formation of electrons, the determination of the semiconductor’s band gap is crucial. For this reason, diffusive reflectance spectroscopy measurements were performed. The reflectance spectrum in the range of 200–1000 nm was only collected for waste and reference samples, which were synthesized at 180 °C for 0.5 and 4 h (Figure 5A). According to XRD results in Figure 2, the samples obtained by the proposed hydrothermal treatment at 145 °C consisted of primary mixture components; therefore, the Sw0.5h145C and Sr0.5h145C samples were not tested. Based on the obtained reflectance spectra, the band gap energy was estimated according to Tauc’s plot [57] (Equations (11)–(13)) (Figure 5B).
(11)$$R=\frac{R_{sample}}{R_{standard}}$$
(12)$$F\left(R\right)=\frac{{\left(1-R\right)}^{2}}{2R}$$
(13)$$F\left(R\right)h\upsilon =A{\left(h\upsilon -E_{g}\right)}^{n}$$
where *R*—the reflectance of sample; *E_g_*—the optical band gap energy (eV); *A*—a proportional constant; *hυ*—the light energy (eV); *n*—a value of depending on nature of electronic transition, e.g., for direct transition *n* = ½, for forbidden direct transition *n* = 3/2, for indirect transition *n* = 2, for forbidden indirect *n* = 3 [57,61]. According to the literature, the value of *n* is 1/2 since the direct transition is characteristic of covellite [29].

According to the linear function of *hυ* to (*F*(*R*)*hυ*)^2^, the determined band gap of the CuS sample obtained from waste during 0.5 h synthesis at 180 °C is 1.78 eV (Figure 5A). It was also observed that the duration of synthesis did not affect the band gap in the waste samples since the value of band gap value in the Sw4h180C sample is 1.78 eV (Figure 5A). On the contrary, the duration of hydrothermal treatment had an influence on the band gap in the reference sample since the value decreased from 1.81 eV to 1.72 eV by increasing the duration from 0.5 to 4 h (Figure 5A). According to literature, the band gap energy characteristic of covellite samples varies around 2.0 eV, showing that the proposed hydrothermal method allowed to obtain particles characterized by experimental bandgap energy slightly lower than expected [47].

According to the obtained band gap values, the calculated electron excitement wavelength varied from 685 to 721 nm; therefore, irradiation visible light stimulates the generation of free electrons in the synthesized samples. In addition, the photocatalytic degradation performance of synthesized CuS samples was evaluated under visible light irradiation and using methylene blue as a model molecule to simulate an organic pollutant. The samples of Sw0.5h180C and Sr0.5h180C were chosen for this experiment. It is worth mentioning that Sw0.5h145C and Sr0.5h145C contained primary mixture components; therefore, the content of photocatalyst in these samples was lower than in Sw0.5h180C and Sr0.5h180C. For this reason, Sw0.5h145C and Sr0.5h145C were not tested in photodegradation experiments. Figure 6A shows the dependence of degradation efficiency as a function of the irradiation time. It was observed that the characteristic absorbance peak at 664 nm of the MB decreased significantly through the first minutes of the experiment (Appendix A). Moreover, the degradation efficiency was lower in the waste sample than in reference by 7% during the primary point of irradiation (Figure 6A). However, by increasing irradiation time up to 30 min, 73% of the dye was degraded with the waste sample, while in the reference sample—61% of MB particles (Figure 6A). At the end of the experiment, after 180 min, the degradation efficiency of both samples was 96% (Figure 6A). The kinetics of degradation reaction of MB, for both photocatalysts, followed the pseudo-first-order kinetics. The rate constant of the first order of the waste sample was 0.0725 min^−1^, while the rate constant of the reference sample was 0.052 min^−1^ (Figure 6B). It is worth mentioning that the obtained reaction rate constants and degradation efficiency rate correspond to the data reported in the literature [28,40].

According to literature, crystallite size has a relevant influence on material properties to degrade organic molecules because the photocatalytic performance improves with increasing crystallite size [62]. In addition, the larger crystallite size of the waste sample determined that the degradation efficiency was higher by 7–20% in the waste sample than in reference during the first half of the experiment. However, it should be underlined that the valorization of the waste sulfurizing agent allowed to obtain particles that have a photocatalytic performance that is comparable with the ones observed for the CuS obtained by conventional sulfurizing agent. In particular, both dye degradation rate and degradation efficiency at 180 min have very similar values for both tested samples.

## 4. Conclusions

A sustainable approach for large-scale copper sulfide (CuS) synthesis that combines the valorization of elemental sulfur-waste, aqueous medium, and mild conditions was herein reported for the first time. Specifically, a hydrothermal and sustainable approach was developed for the direct synthesis of CuS by using copper oxide and sulfur-based by-products of sulfuric acid production as a sulfuring agent. It is noteworthy that the obtained high-value CuS particles have been then characterized, and their properties have been compared to reference particles obtained by the same procedure but using a conventional sulfuring agent.

The experimental results and thermodynamic calculations proved that the initial reaction of covellite synthesis was the interaction of melted sulfur and water to form hydrogen sulfide gas and sulfur dioxide in the soluble state. This reaction initiated the formation of hydrogen cations and hydrogen sulfide anions, which, together with the H_2_S gas, further interacted with the copper sulfide resulting in the desired target product.

Furthermore, the average crystallite size of synthetic covellite varied from 27 to 40 nm depending on the synthesis conditions (145–180 °C, 0.5–4 h), but no significant variation was observed when the sulfur source was changed. Furthermore, the band gap energy has not significantly varied when the waste sulfuring agent was used for the synthesis. In particular, the bandgap of the synthesized CuS particles is in the range of 1.72–1.81 eV, thus making these particles suitable for photocatalytic degradation of organic pollutants activated by visible light. In detail, the degradation of methylene blue of the sample that has been synthesized from waste valorization showed a remarkable degradation efficiency of 96 % in 3 h, following the first-order kinetics. The degradation rate of the waste and reference samples were 0.052 and 0.0725 min^−1^, respectively, showing once again that the proposed approach allowed to obtain photocatalysts with properties that are comparable with the reference samples.

The presented approach for the synthesis of CuS from sulfur waste can be successfully applied for environmentally friendly large-scale production of photocatalytic covellite with potential application for water remediation by visible light activation.

## Figures and Tables

**Figure 1 materials-15-05253-f001:**
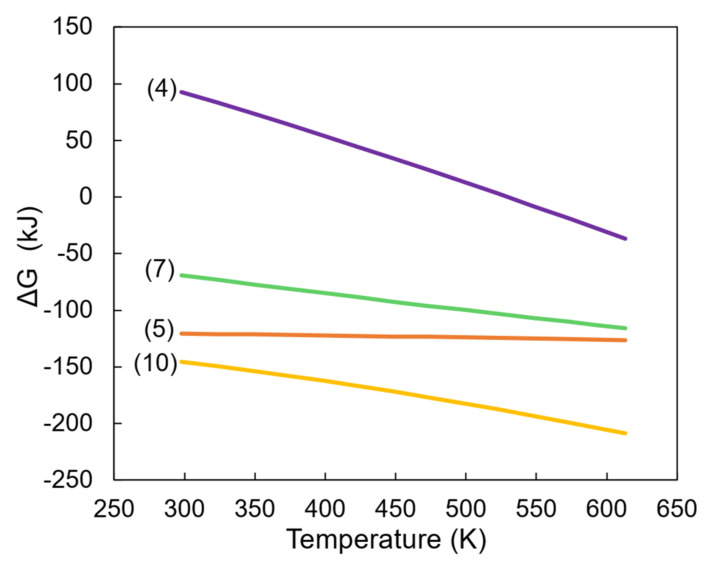
The dependence of Gibbs free energy on temperature.

**Figure 2 materials-15-05253-f002:**
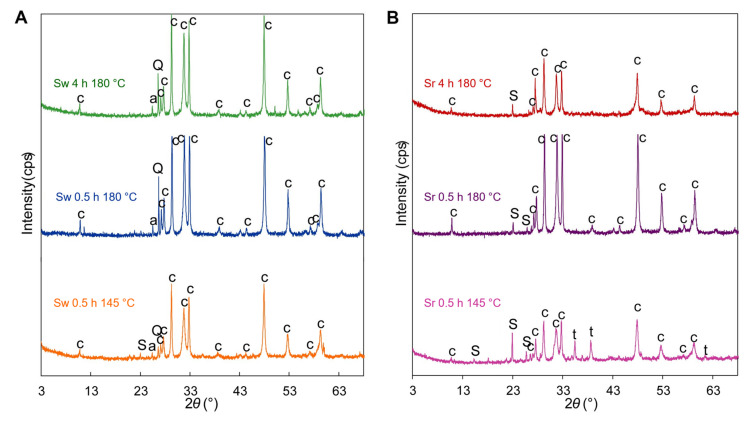
XRD curves of hydrothermal synthesis products. (**A**)—sulfur waste samples; (**B**)—sulfur reference samples. S—sulfur; c—covellite; Q—quartz; t—tenorite; a—anhydrite.

**Figure 3 materials-15-05253-f003:**
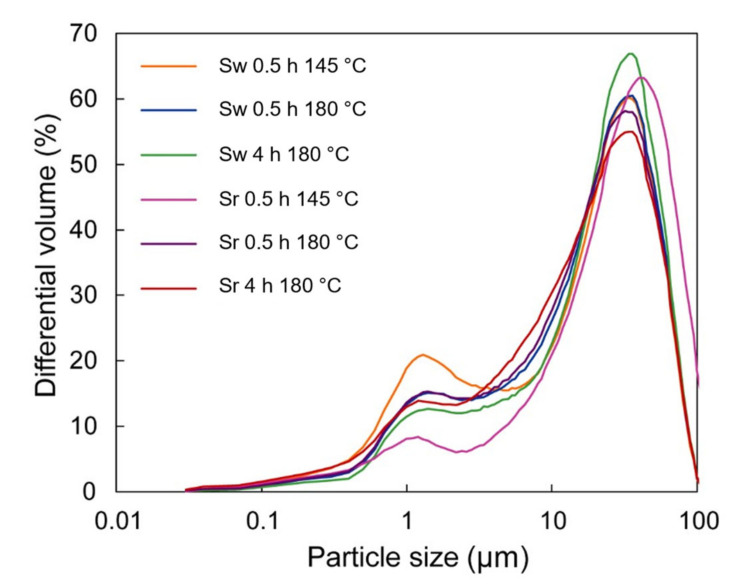
Particle size distribution of synthesis products.

**Figure 4 materials-15-05253-f004:**
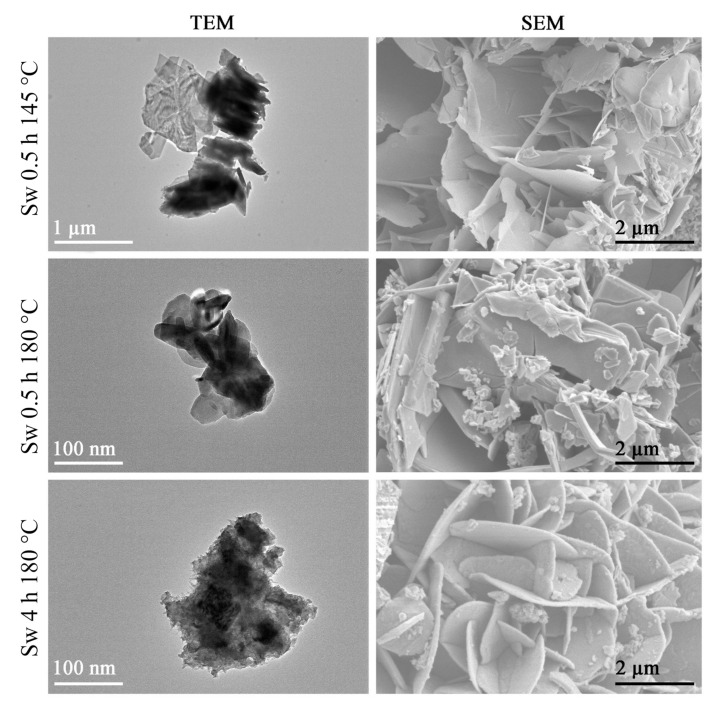
SEM and TEM images of synthesized copper sulfide samples from sulfur waste.

**Figure 5 materials-15-05253-f005:**
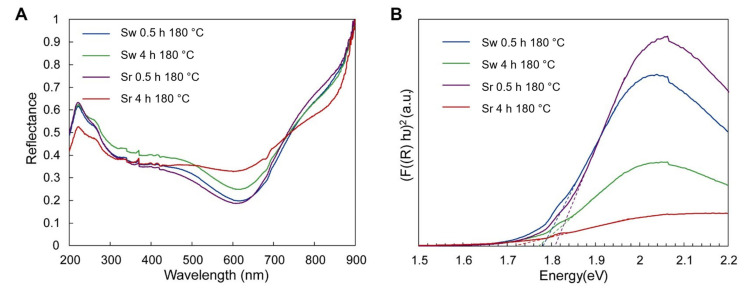
(**A**) diffusive Reflectance spectra of samples; (**B**) Kubelka–Munk function plot of synthesized CuS waste and reference samples.

**Figure 6 materials-15-05253-f006:**
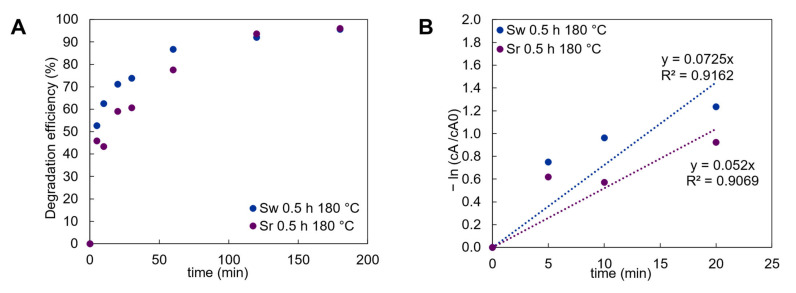
The data of photocatalytic activity of synthesized products in a MB and H_2_O_2_ solution. (**A**)—photocatalytic degradation efficiency; (**B**)—the plot of first order kinetics. Black—Sw0.5h180C; orange—Sr0.5h180C.

**Table 1 materials-15-05253-t001:** Experimental conditions.

Sulfurizing Agent	Duration (h)	Temperature (°C)	Abbreviation
Sulfur waste	0.5	145	Sw 0.5 h 145 °C
		180	Sw 0.5 h 180 °C
	4		Sw 4 h 180 °C
Sulfur reference	0.5	145	Sr 0.5 h 145 °C
		180	Sr 0.5 h 180 °C
	4		Sr 4 h 180 °C

**Table 2 materials-15-05253-t002:** Standard molar thermodynamic properties at 25 °C and 1 bar *****.

Component	$\Delta H_{298}^{0,f}\;(\mathrm{kJ}\cdot {\mathrm{mol}}^{-1})$	$\Delta S_{298}^{0,f}\;(J\cdot {\mathrm{mol}}^{-1}\cdot K^{-1})$	$\Delta C_{p}\;(J\cdot {\mathrm{mol}}^{-1}\cdot K^{-1})$
H^+^ (aq)	0	0	0
H_2_O (l)	–285.8	70.0	75.3
S (l)	1.4	35.2	23.6
SO_2_ (aq)	–323.8	159.5	195.0
H_2_S (g)	–20.6	205.8	34.2
HS^−^ (aq)	–17.6	62.8	NG
Cu^2+^ (aq)	64.8	–99.6	NG
CuO (s)	–157.3	42.6	42.3
CuS (s)	–79.5	120.9	76.3

* aqueous (aq), liquid (l), gas (g), solid (s).

**Table 3 materials-15-05253-t003:** Crystallite size of synthesized covellite.

Sample	Waste	Reference
	Crystallite Size ± Standard Dev. (nm)	
0.5 h 145 °C	32.8 ± 9.5	27.2 ± 10.5
0.5 h 180 °C	38.7 ± 9.4	34.8 ± 10.6
4 h 180 °C	39.5 ± 9.3	36.6 ± 9.7

**Table 4 materials-15-05253-t004:** The average quantification of EDX spectra of different sulfur waste samples.

Sample	*Atomic Ratio Cu:S*
Sw 0.5 h 145 °C	1.01:1
Sw 0.5 h 180 °C	1.06:1
Sw 4 h 180 °C	0.87:1

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
