# Peer review of "Aqueous-Based Synthesis of Photocatalytic Copper Sulfide Using Sulfur Waste as Sulfurizing Agent"

_materials, 2022, doi:10.3390/ma15155253_

Round 1
Reviewer 1 Report
The manuscript entitled: 'Aqueous – based synthesis of photocatalytic copper sulfide using sulfur waste as sulfurizing agent' reports a new way of synthesizing copper sulfide. The manuscript has several shortcomings that should be addressed before publication. Overall, it is well written, however the discussion in the whole manuscript is very poor and in some section it is not present. Authors should discuss the results and compare them to the literature.
Follow a detailed review:
Line 37: ‘is considered to be an abundant nontoxic’. This sentence should be reformulate) https://www.sciencedirect.com/science/article/pii/S1549963415000210?via%3Dihub)
Line 55: ‘Among them, hydrothermal synthesis is one of the most employed techniques due to process advantages, such as low production cost, easy processing, eco-friendliness [39–42]’, Line 61: ‘Furthermore, the majority of copper sulfide synthesis are carried out in organic reagents as reaction medium, which are environmentally unsustainable’ and Line 66: ‘In addition, copper sulfide production costs are typically very high’. These sentences are in contradiction. The authors should better explain what they meant.
Line 73: ‘However, few studies have investigated the application of cheap and earth abundant reagents, for instance insoluble copper or sulfur sources’ Citation should be added.
Line 145: The authors in Eq. 2 used absorbance, while in Eq. 3 used concentration. An explanation is needed. The reviewer suggested using the concentration also in Eq. 2
Line 149: ‘(mg/l)’ line 131 reports mg‧l -1. Authors should unify the units
Line 236 : ’(Fig)’. It should be Fig 3
Line 236: ‘The majority of all samples (53.7-70.1 %) consisted of particles less than 30 μm (Fig. 3).’ In line 234: ‘It was determined that mainly all the samples consisted of particles less than 100 μm’ Authors should better explain.
Line 234: ‘The particle size of bulk synthesized material was further investigated’. How was the size distribution calculated?
Line 267: ‘Fig. 4. SEM and TEM images of synthesis products’. The caption needs to be improved.
Line 328: Authors should also report blank tests.
11) MB+ H2O2 without catalyst in the dark,
22) MB+H2O2 without catalyst under light irradiation,
33) MB+ catalyst in the dark,
44) MB+ H2O2 + catalyst in the dark,
55) MB+H2O2 under light irradiation
In addition, there is also no comparison with other photocatalysts reported in literature.
Round 2
Reviewer 1 Report
The authors responded point by point to all comments. The manuscript should be considered for publication.